# Cancer-Associated Fibroblasts in Pancreatic Ductal Adenocarcinoma: An Update on Heterogeneity and Therapeutic Targeting

**DOI:** 10.3390/ijms222413408

**Published:** 2021-12-14

**Authors:** Utpreksha Vaish, Tejeshwar Jain, Abhi C. Are, Vikas Dudeja

**Affiliations:** Department of Surgery, University of Alabama at Birmingham, Birmingham, AL 35233, USA; utprekshavaish@uabmc.edu (U.V.); tjain@uabmc.edu (T.J.); acare2019@hotmail.com (A.C.A.)

**Keywords:** pancreatic ductal adenocarcinoma, pancreatic cancer, cancer-associated fibroblasts, myCAF, iCAF, apCAF, CAF heterogeneity

## Abstract

Pancreatic ductal adenocarcinoma (PDAC) is a leading cause of cancer-related morbidity and mortality in the western world, with limited therapeutic strategies and dismal long-term survival. Cancer-associated fibroblasts (CAFs) are key components of the pancreatic tumor microenvironment, maintaining the extracellular matrix, while also being involved in intricate crosstalk with cancer cells and infiltrating immunocytes. Therefore, they are potential targets for developing therapeutic strategies against PDAC. However, recent studies have demonstrated significant heterogeneity in CAFs with respect to their origins, spatial distribution, and functional phenotypes within the PDAC tumor microenvironment. Therefore, it is imperative to understand and delineate this heterogeneity prior to targeting CAFs for PDAC therapy.

## 1. Introduction

Pancreatic cancer is a lethal malignancy. Five-year survival of patients with pancreatic cancer has only recently reached double digits (~10%) [1,2], with pancreatic cancer projected to become the second leading cause of cancer-related deaths in the United States by 2030 [3]. About 85% of all pancreatic cancer cases are characterized as pancreatic ductal adenocarcinoma (PDAC) [4,5]. Surgery remains the only curative option for potentially resectable patients, while patients with more advanced disease are usually treated with systemic therapies, such as chemotherapy [6,7]. Although the standard-of-care chemotherapy regimens do provide survival benefits, the overall survival still remains dismal. Aggressive biology and resistance to therapy are some of the factors that contribute to these poor outcomes [6]. Recent studies have demonstrated that, apart from the cancerous epithelial compartment, other cellular and non-cellular elements present in the tumor microenvironment (TME) of PDAC also contribute substantially to its aggressive biology, progression, and metastasis [8,9]. Therefore, PDAC TME has been extensively investigated as a target to modulate PDAC progression, and to enhance the efficacy of currently available anti-cancer therapies.

## 2. The PDAC Tumor Microenvironment

PDAC tumorigenesis involves progressive accumulation of mutations in driver genes, such as *KRAS*, *CDKN2A*, *SMAD4*, and *TP53*, which can be accompanied by various passenger mutations [6]. This leads to the development of precursor lesions, such as pancreatic intraepithelial neoplasia (PanINs), which transform into invasive lesions as the disease progresses. To support their proliferation, cancer cells hijack the transcriptional machinery of the surrounding stroma, and create a highly specialized niche conducive to cancer growth, known as the tumor microenvironment (TME). It is characterized by aberrant activation of numerous signaling pathways, including growth factor signaling, angiogenesis, metabolism, etc. [6]. Histological analysis of resected PDAC specimens reveals a highly desmoplastic stroma with poor vascularity. Cellular components of the TME chiefly consist of cancer-associated fibroblasts (CAFs) and immune cells of both lymphoid and myeloid lineages, with sparse representation from other cell types, including endothelial cells and neural cells [10]. These members of the TME interact with the cancer cells, as well as with each other, to support cancer growth. Indeed, preclinical and clinical studies suggest that components of the TME might serve as viable therapeutic targets against PDAC. Researchers have successfully targeted various subsets of lymphoid [11,12] and myeloid immune cells [13,14,15] to affect tumor growth and metastasis. Furthermore, TME targeting also has the potential for overcoming resistance to immune-checkpoint blockade therapy in PDAC, possibly opening a new avenue for cancer therapy [9,16]. In the following sections, we summarize the current literature on the significance of CAFs, the dominant cell type in the PDAC TME, and various targeting strategies that have been employed to modulate CAFs for cancer therapy.

## 3. Cancer-Associated Fibroblasts (CAFs)

Fibroblasts are tissue-resident, spindle-shaped cells that primarily arise from the primitive mesenchyme, with the neural crest giving rise to a minor population, as well [17,18]. They are present in the connective tissue of nearly all solid organs and play a vital role in tissue repair, converting from a quiescent to an activated contractile myofibroblastic type, to maintain tissue homeostasis [19]. Fibroblasts secrete a wide variety of extracellular matrix (ECM) components, such as collagens, proteoglycans, and growth factors; proteolytic enzymes, such as matrix metalloproteinases (MMPs), and small signaling molecules, such as cytokines, and therefore, fibroblasts have the ability to remodel the ECM. Cellular elements with fibroblast-like histology and functional features can be identified throughout various solid tumors, and thus, have been annotated as cancer-associated fibroblasts (CAFs).

CAFs are the major non-neoplastic component of the tumor microenvironment in PDAC. CAFs not only contribute to the desmoplastic stroma by secreting extracellular matrix proteins, but also modulate cancer progression by influencing the production of cytokines/chemokines, such as transforming growth factor-β (TGF-β), vascular endothelial growth factor (VEGF), interleukin-6 (IL-6), and CXC chemokine ligand 12 (CXCL12) [20,21]. Moreover, CAFs have also been reported to play an integral role in immune evasion and poor responses to immunotherapy [22]. These characteristics make CAFs potential therapeutic targets for treating pancreatic cancer, where despite decades of research, efficacy of existing chemo- and immunotherapies remains thoroughly underwhelming.

## 4. Origin of CAFs

Although CAFs have been extensively evaluated using various preclinical models [23], it has been a challenge to characterize them well, especially in the context of their lineage, primarily due to the unavailability of exclusive cell markers [23]. CAFs are primarily defined as cells that lack lineage markers for epithelial cells (Epithelial Cell Adhesion Molecule or EpCAM or CD326), endothelial cells (CD31), and leukocytes (CD45); possess elongated, spindle-shaped morphology; and lack the mutations found within the cancer cells [24]. A significant proportion of CAFs has been discovered to arise from the resident tissue fibroblasts [25,26]. Resident fibroblasts can be triggered by the signals released from neighboring tumor cells to form CAFs. However, besides fibroblasts, mesenchymal stem cells (MSCs) [27,28], mesothelial cells (through mesothelial–mesenchymal transition or MMT) [29], adipocytes [30], hematopoietic stem cells [31], and circulating bone marrow cells called fibrocytes [32], have also been shown to transform into CAFs (Figure 1). Epithelial cells [33] and endothelial cells [34] can transform into fibroblasts through epithelial-to-mesenchymal transition (EMT) or endothelial-to-mesenchymal transition (EndMT), respectively, and these fibroblasts have been suggested as sources of CAFs, as well. Iwano et al. showed that, during renal fibrogenesis, fibroblast-specific protein^+^ (FSP^+^) and CD34^−^ fibroblasts arise from bone marrow and migrate to the renal interstitial space, but a large number of FSP^+^ fibroblasts are contributed by the local epithelial–mesenchymal transition [33]. Zeisberg et al. discovered that transforming growth factor-beta 1 (TGF-β1) could drive the conversion of lung endothelial cells into fibroblast-like cells by EndMT and contribute to the pool of CAFs [34]. Quante et al. have performed bone marrow reconstitution studies in inflammation-induced, gastric cancer mouse models, and found that approximately 20% of CAFs are known to arise from the bone-marrow-derived mesenchymal stem cells [35]. Kidd et al. have studied generation of adipocyte-derived CAFs in a murine breast-cancer model using GFP^+^ adipocytes, and found that α-smooth muscle actin or (αSMA^+^) tumor stroma were generated from the local adipose tissue [30]. Jotzu et al. have shown that a significant percentage of human adipose-tissue-derived stem cells (hASCs) differentiated into CAFs when exposed to a human breast-cancer-conditioned medium containing significant amounts of TGFβ1. Their group also suggested that the differentiation of hASCs into CAFs is dependent on TGFβ1 signaling via Smad3 [36]. However, lineage-tracing studies to identify specific CAF progenitors remain encumbered by the lack of CAF-restricted markers.

With regard to PDAC, pancreatic stellate cells (PSCs), the resident fibroblasts of the pancreas, were long presumed to be the putative precursor cells for CAFs. Functional characterization of PSCs was first undertaken when activated PSCs (expressing αSMA) were demonstrated to be the chief sources of fibrosis in chronic pancreatitis [37]. Interestingly, immunostaining studies showed a significant correlation between αSMA and collagen in the desmoplastic regions in human pancreatic cancer specimens [38]. Using the outgrowth method, Bachem et al. isolated αSMA^+^ pancreatic stellate cells from human PDAC and chronic pancreatitis specimens, and showed that they were morphologically alike with similar staining for markers, such as collagen, fibronectin, and desmin. Moreover, multiple studies found that cancer-cell-conditioned media could stimulate PSCs to proliferate and produce ECM proteins [37,39,40]. Tumors formed on co-injection of cancer cells with PSCs also showed more intense desmoplasia with a greater number of αSMA^+^ cells [39]. Such studies supported the notion that PSCs were the dominant sources of CAFs in PDAC. However, accumulating evidences have started to challenge this long-held notion.

Helms et al. have recently performed fate-mapping studies on quiescent PSCs during pancreatic carcinogenesis. Using fatty acid-binding protein 4 (Fabp4) as a maker for quiescent PSCs, they generated an elegant dual-labelled mouse model, where Fabp4 expression drove GFP labelling of PSCs, while the rest of the cells in the body were ubiquitously labelled with tdTomato [41]. Interestingly, they found that PSCs only gave rise to about 15% of the CAF population in the PDAC TME, which would again indicate a greater variability in CAF precursors. This hypothesis is in concordance with the results of Dominguez et al. who employed a combination of bulk and single-cell RNA sequencing (RNAseq) on normal and malignant pancreatic tissue of human and murine origin, to trace the evolution of resident fibroblasts in normal tissues into CAFs, as pancreatic cancer develops [42]. Their studies revealed the presence of a heterogenous fibroblast population in normal pancreatic tissue. Interestingly, these populations appeared to evolve into different CAF subsets as the tumors progressed, when the authors performed trajectory analysis on their single-cell RNAseq data. A fibroblast population characterized by expression of elastin fibrils and ECM attachment proteins gave rise to CAFs with a TGFβ-driven transcriptional program, while another fibroblast population, characterized by collagen expression, gave rise to an interleukin-1 (IL-1) driven CAF population [42]. These findings indicate that CAF heterogeneity might be predetermined, based on the type of progenitor cell from which CAFs arise. Garcia et al. further shed light on the heterogenous fibroblastic precursors of CAFs in PDAC when they performed lineage-tracing studies using *Gli1* and *Hoxb6*, two mesenchymal markers that have not been associated with stellate cells [43]. Both Gli1 and Hoxb6 were found to be expressed in cells expressing fibroblast markers, such as αSMA and platelet derived growth factor receptor α (PDGFRα); however, they appeared to mark distinct fibroblast populations with minimal overlap. Interestingly, fate mapping of these cells in genetically engineered mouse models (GEMMs) of PDAC revealed that only the Gli1^+^ fibroblasts expanded during carcinogenesis and contributed to the αSMA^+^ CAF pool, while the Hoxb6^+^ fibroblasts did not appear to give rise to CAFs [43]. There is also evidence stemming from pancreatic cancer cell-implantation experiments that seems to support a contribution from cells, such as mesenchymal stem cells [28,44,45] to the CAF pool in pancreatic cancer, although definitive evidence using lineage-tracing studies is lacking in this regard. Additionally, there have been reports of cells with CAF-like features isolated from the TME of pancreatic cancer GEMMs, which possess abundant mesothelial markers on transcriptional analysis, indicating a possible mesothelial origin, as well [46,47]. Thus, further investigations using lineage-tracing studies are warranted to accurately establish the identities of potential CAF precursors, and the functional relevance that these origins might imply.

## 5. CAF Heterogeneity

Since the tumor microenvironment was looked upon as an active player, rather than a bystander, in tumor progression and metastasis, targeting of stroma and CAFs has been evaluated as a strategy to improve the PDAC outcome. However, the results have been conflicting [48,49]. Early results indicating a protumorigenic function of CAFs came from studies where PSCs, which are known to differentiate into CAFs in the PDAC TME, were co-injected with KPC (*KrasLSL-G12D*; *Trp53 LSL-R172H/+*; *Pdx1Cre/+)* cancer cells orthotopically into the tail of the pancreas. These co-injections led to significantly larger tumors, as compared to injections of cancer cells alone [50]. This resulted in multiple studies targeting CAFs, using specific markers or signaling pathways for pancreatic cancer therapy. One such marker, which was extensively used in the initial studies, was the fibroblast activation protein-α (FAPα). Kraman et al. found that depletion of FAP-expressing stromal cells in a subcutaneous model of PDAC reduced tumor burden by instigating rapid hypoxic necrosis of both cancer cells and stromal cells in an interferon-γ (IFN-γ) and tumor necrosis factor-α (TNF-α) dependent manner [48]. However, in their model, FAP^+^ cells in the PDAC TME consisted of both CD45^+^ and CD45^−^ fractions, suggesting that this effect was not limited to the CAF population. In a separate study, Feig et al. found that the majority of FAP^+^ cells in the KPC GEMM of PDAC were αSMA^+^, CD45^−^, CD34^−^, and PDGRα^−^, leading them to conclude that FAP could be a specific marker for CAFs in the PDAC TME. DTR-mediated depletion of FAP-positive cells in an implanted model of pancreatic cancer decreased tumor burden and relieved immunosuppression [21]. Therefore, it appeared that CAFs were tumor-promoting entities that could be depleted to provide a therapeutic benefit. Surprisingly, Ozdemir et al. observed that the depletion of αSMA^+^ CAFs in pancreatic cancer led to invasive, undifferentiated tumors with diminished animal survival, and also increased the number of cancer stem cells [49]. Rhim et al. obtained similar results when they attempted deletion of sonic hedgehog (Shh), one of the drivers of desmoplastic stroma, in mouse models of PDAC. This approach led to reduced stromal content, but aggressive and undifferentiated tumors, indicating a possible tumor-restraining role of the stromal desmoplasia [51]. These conflicting findings forced a systematic review of the presumed PDAC stromal model, consisting of homogenous CAF populations that support tumor growth. Recent studies have begun to uncover a more diverse transcriptional program in the PDAC CAF populations, leading to identification of distinct subtypes with functional plasticity, resulting in a biologically dynamic phenotype (Figure 2).

Öhlund et al. used a three-dimensional, in vitro co-culture model system, consisting of pancreatic stellate cells (PSCs) and KPC (*KrasLSL-G12D*; *Trp53 LSL-R172H/+*; *Pdx1Cre/+)* mouse-derived PDAC organoids to decipher CAF heterogeneity in PDAC [40]. In this model, the authors identified two types of CAFs, namely, “myofibroblastic CAFs” (myCAFs) and “inflammatory CAFs” (iCAFs). The myCAFs possess a contractile phenotype, and are generally characterized by high expression of αSMA (*ACTA2*) with an ECM signature, whereas iCAFs possess an inflammatory phenotype, with a high expression of interleukin-6 (*IL-6*) and a low expression of αSMA (*ACTA2*). These CAFs are spatially divergent as well, i.e., myCAFs have been found to be proximal to the cancer cells, whereas iCAFs are distant from the cancer cells [40]. It appears that this dichotomy in CAF phenotypes arises as a result of the cues coming from the neoplastic compartment. These cues were further explored by Biffi et al. and it appears that the secretion of TGFβ and IL-1 from the cancer cells contribute to the CAF heterogeneity [52]. The authors demonstrated that IL-1 acts through NF-κB and IL-6, induces expression of the leukemia inhibitory factor (LIF), and activates downstream Janus kinase/signal transducer and activator of transcription (JAK/STAT) to generate inflammatory CAFs; and conversely, TGFβ antagonizes this process by suppression of the IL-1 receptor, IL1R1, and promotes differentiation into myofibroblasts. In summary, IL1/JAK–STAT3 and TGFβ/SMAD2/3 are two opposing signaling pathways that induce iCAF or myCAF formation, respectively. It is also evident that these states are plastic, and that these two forms of CAFs are interconvertible in vitro, i.e., iCAFs cultured in Matrigel with a PDAC organoid or its conditioned medium will convert to myofibroblasts, if cultured in a two-dimensional monolayer. Additionally, a small proportion of αSMA/pSTAT3 double-positive CAFs were also identified, which might represent a transitional state between the iCAF and myCAF phenotypes, further supporting the plasticity between these two CAF populations in vivo [52]. The cell-specific transcriptional signature indicates that myCAFs contribute to desmoplastic stroma in the tumor microenvironment, whereas iCAFs are sources of inflammatory cytokines. Inhibiting the IL1/NF-κB-mediated signaling pathway using JAK inhibitors led to decreased tumor growth, accompanied by conversion of iCAFs into myCAFs, indicating that the iCAF phenotype might possess tumor-promoting properties [52]. On the other hand, myCAFs might possess a tumor-restraining role, as evidenced by increased aggressiveness of tumors upon targeting CAFs resembling this subtype. More recently, another study investigating the effects of regulatory T cells (Tregs) depletion on pancreatic cancer precursor-lesion progression demonstrated rapid growth and poor differentiation of PDAC tumors upon targeting CD4^+^ Foxp3^+^ T cells (Tregs) [53]. This surprising observation seemed to be a consequence of reprograming the fibroblast population, with decreased αSMA+ myofibroblasts. These findings indicate a possible interplay between the immune and CAF compartments to restrain tumor growth during early tumorigenesis, highlighting the complex role that CAFs play in the tumor microenvironment.

To investigate the neoplastic and tumor microenvironment content of human and murine PDAC tumors in detail, Elyada et al. utilized a droplet-based, single-cell RNA sequencing (scRNAseq) approach. Furthermore, the authors also enriched the fibroblasts prior to transcriptome profiling to investigate CAF heterogeneity thoroughly. Besides corroborating the presence of myCAFs and iCAFs in both species, the authors identified a new population of CAFs that express major histocompatibility complex class II (MHC II) and CD74, termed as “antigen-presenting CAFs” or apCAFs. This was the first instance of fibroblasts with antigen-presenting capabilities being identified in a neoplastic environment. The apCAFs also showed higher activity for STAT1, which is known to mediate MHCII expression in response to IFNγ, suggesting that apCAFs are regulated by IFNγ signaling in vivo [46]. The authors also evaluated the ability of apCAFs to activate CD4^+^ T cells in an antigen-specific manner ex vivo. Although apCAFs did possess a limited ability to activate CD4+ T cells, unlike professional antigen-presenting cells (APCs), they did not express classic costimulatory molecules, such as CD40, CD80, or CD86, suggesting that apCAFs in PDAC may play a different role as compared to the professional APCs. This led the authors to hypothesize that the MHCII expressed by apCAFs may act as a decoy receptor to disengage the CD4+ T cells, preventing their clonal proliferation, and thus, leading to T-cell anergy or differentiation into Tregs, contributing to an immunosuppressive TME. Importantly, the authors were able to demonstrate flow-cytometry-validated cell-surface markers for isolation of different CAF subtypes from human and murine PDAC lesions, which could serve as a valuable tool for future studies aiming to characterize these subsets, and to evaluate their role in tumor progression and response to therapy. This report also identified podoplanin (*PDPN*) and decorin (*DCN*) as pan-CAF markers in all three CAF populations, i.e., myCAFs, iCAFs, and apCAFs, which can be further investigated to be utilized as exclusive CAF markers.

A recent study by Hutton et al. has made progress towards identification of possible lineage markers for functionally distinct CAF subtypes in PDAC [47]. Using stringent mass cytometry panels to isolate mesenchymal populations from spontaneous murine PDAC tumors, they utilized single-cell transcriptomics to characterize the stromal heterogeneity. The authors identified CD105, a membrane glycoprotein that is a part of the TGFβ receptor complex, as a stable marker that could reliably distinguish transcriptionally different subsets of CAFs, as well as normal tissue fibroblasts, a finding that was reaffirmed on analysis of a single-cell RNA-seq dataset of human PDAC specimens. Notably, both CD105^+^ and CD105^−^ CAFs displayed expression of genes associated with previously described myCAF and iCAF subgroups, although the apCAF expression profile seemed to be restricted to a subset of CD105^−^ CAFs. The authors also found that single-cell RNA sequencing was unable to completely capture *Eng* (the gene coding for CD105) expression even in CD105^+^ cells, indicating that transcriptomics alone cannot clearly distinguish heterogenous stromal populations. Interestingly, CD105^−^ CAFs appeared to be tumor-restricting in vivo, and this effect seemed to depend on a functional adaptive immune system, as well as on cDC1 dendritic cells.

Another interesting aspect of CAF heterogeneity is their discrete spatial localization in the PDAC microenvironment. Grünwald et al. [54] characterized this heterogeneity in the context of overall stromal heterogeneity in resected human PDAC specimens, as well as in biopsies from metastatic PDAC lesions. The authors classified the PDAC TME into two main, histologically distinguishable entities—the ‘Deserted’ TME, with an acellular appearance and thin, spindle-shaped fibroblasts, and the ‘Reactive’ TME, with active fibroblasts and abundant inflammatory infiltrate; an intermediate TME state between these two could also be identified. Combined proteomics and transcriptomics analysis on microdissected TME samples demonstrated a significantly different gene and protein expression signature between these TME variants (labelled subTMEs by the authors). Interestingly, CAFs isolated from these subTMEs also possessed distinct transcriptional signatures and functional phenotypes. In vitro, CAFs from deserted TMEs conferred chemoresistance to cancer cells, while CAFs from the reactive TME supported proliferation of the basal/squamous-type cancer cells. Although the transcriptional signature of these CAF populations did not align along the myCAF–iCAF axis, distinct transcriptional programs could still be identified, with deserted TME CAFs enriched in cell cycle genes, and the reactive TME CAFs showing upregulation of inflammation-associated genes. Overall, these findings indicate that CAFs may organize as parts of discrete sub-stromal niches with distinct tumor-supportive functions.

As more and more evidence come to light, it has become increasingly apparent that the scope of CAF diversity is much greater than previously appreciated, and this calls for more investigations to understand CAF subpopulations, their functions, and their role in tumor progression. Furthermore, it will be important to delineate the contribution of the different sources of CAFs to this functional heterogeneity, as recent studies seem to indicate that CAFs with distinct origins might have nonredundant functional phenotypes [41,46]. These studies will be essential as we attempt to engage CAFs for developing novel chemo- and immunotherapeutic strategies against PDAC.

## 6. Functions of CAFs

CAFs mediate a variety of effects in the TME and can promote tumor growth by multiple mechanisms (Figure 3). As discussed before, PDAC is characterized by an extensive desmoplastic reaction caused by the CAF-mediated deposition of an ECM rich in collagens (*COL1A1*, *COL1A2*), glycoproteins, and proteoglycans. ECM acts as a barrier to drug delivery, supports tumor growth by providing biochemical cues, contributes to immunosuppressive TME [55], and enhances expression of matrix metalloproteinases (MMPs), which facilitate cell invasion and metastasis [56]. Matrix crosslinking enzymes and ECM remodeling also contribute to the increased stiffness of tumor tissue, which can cause hypoxia and a more aggressive cancer phenotype [57]. It also restricts drug delivery and triggers pro-survival signaling in cancer cells [58]. As the cancer metastasizes to secondary sites, CAFs favor the establishment of cancer via production of ECM components, facilitating immune exclusion, and providing survival cues to cancer cells [59].

Tumor growth and progression are largely dependent on angiogenesis that is well supported by CAFs through the release of vascular endothelial growth factor (VEGF) [60]. Since the tumor microenvironment is hypoxic in nature, it further upregulates expression of *VEGF* by CAFs. IL-6 released by CAFs is also known to promote angiogenesis and helps in tumor progression [61]. CAFs can also provide metabolic support for cancer cells during tumorigenesis. Zhang et al. have reported that glutamine deficiency triggers macropinocytic nutrient uptake in pancreatic CAFs by the enhancement of cytosolic Ca2^+^ and CaMKK2-AMPK signaling. Herein, macropinocytosis contributes to intracellular and extracellular amino acid pools that sustain CAF fitness and promote tumor cell survival [62].

CAFs secrete a battery of growth factors and cytokines that can promote tumor growth and modulate the response to therapy. CAFs have been reported to establish an immunosuppressive environment by secreting biomolecules, such as IL-6, CXCL12, TGF-β, growth arrest-specific protein 6 (GAS6), fibroblast growth factor 5 (FGF5), growth differentiation factor 15 (GDF15), and hepatocyte growth factor (HGF), which promotes invasive and proliferative behavior in cancer cells [63,64,65]. CAFs also have immunosuppressive effects on immune cells, such as CD8+ T cells, Tregs, and macrophages [55] caused via IL-6, CXC chemokine ligand 9 (CXCL9), and TGFβ [66]. CAFs also prevent CD8 T-cell infiltration in tumors [67], and they recruit immunosuppressive cell populations, such as myeloid-derived suppressor cells (MDSCs) and neutrophils [55]. IL-6 may also promote immunosuppression via systemic effects on metabolism [68].

## 7. CAFs as Therapeutic Targets

The critical and diverse functions performed by CAFs in the tumor microenvironment make them attractive targets for antitumor therapies. Modulation of the tumor microenvironment through direct targeting/depletion of CAFs, or disrupting their cross-talk with cancer cells and immune cells, has been attempted with varying degrees of success in both preclinical studies and clinical trials (Table 1). These studies have also provided insights into mechanisms by which CAFs promote tumor growth.

Depletion of CAFs in the TME: Guided by the belief that CAFs help to create an environment conducive for tumorigenesis, researchers initially attempted direct depletion of CAFs to constrain tumor growth. One such approach involved targeting αSMA^+^ cells in a PKT model *(KrasLSL-G12D*; *Tgbfr2flox/flox*; *Ptf1aCre/+)* of PDAC. Contrary to expectations, this approach resulted in a more aggressive phenotype upon myofibroblast depletion, accompanied by increased Foxp3^+^ Treg infiltration, and consequent immunosuppression [49]. A subsequent study showed that collagen type 1 deposition by these αSMA^+^ CAFs is essential for restraining tumor growth, as specific deletion of *Col1* leads to CXC chemokine ligand 5 (*CXCL5)* upregulation in cancer cells, which can attract immunosuppressive MDSCs into the PDAC microenvironment [69]. These results further reinforce the heterogeneity and distinct functional profiles of CAFs in the PDAC TME, as certain populations do seem to have a tumor-restraining role. Another approach attempting to target CAF markers is based on FAP (fibroblast activation protein) inhibition. Genetic deletion of FAP in a KPC model led to decreased tumor growth, with improved immune surveillance and synergy with anti PD-1 and anti CTLA-4 therapies, an effect that resulted from a reduction in CAF-mediated CXCL12 secretion. A separate study found that FAP deletion also led to decreased tumor invasion and metastasis in preclinical PDAC models [70]. However, given that FAP inhibitors had underwhelming results in clinical trials with metastatic colorectal cancer patients [71], direct targeting of FAP^+^ cells were not pursued in PDAC. Interestingly, recent studies have adopted immunotherapy-based approaches in targeting FAP^+^ CAFs, with encouraging results in pancreatic, colon, and lung cancers [72,73,74]. Experiences with therapies directed against cell-specific targets, such as αSMA and FAP, indicate that there is a need to further characterize CAF-specific markers, and to define the functional significance of such subtypes before attempting targeted therapies. Therefore, instead of attempting direct depletion, researchers have also tried to focus on the functional aspects of CAFs in the PDAC microenvironment.

Targeting ECM production by CAFs: CAFs are major producers of ECM components, such as collagen and proteoglycans. Moreover, they can release proteolytic enzymes, which instigate ECM remodeling and can facilitate tumor invasion and metastasis. The desmoplasia also serves as a major barrier to effective drug delivery, and serves to exclude antitumor immune cells from the PDAC TME. Sonic hedgehog (Shh) signaling, which is essential for the embryonic development of the pancreas, has been extensively investigated in this regard. Shh overexpression can drive desmoplasia in PDAC [75]. However, studies targeting the Shh pathway have shown contrasting results. Administration of Shh inhibitors can lead to decreased tumor growth, decreased desmoplasia, and increased vascularity and drug delivery in genetic models of PDAC [76,77]. However, when Shh is specifically deleted in cancer cells, more aggressive and undifferentiated tumors are formed with reduced animal survival [78]. This is accompanied by reduced Shh signaling in the accompanying fibroblasts, suggesting that this approach might interfere with the tumor-restraining myofibroblasts [78]. One possible explanation leveraged to explain these contrasting roles is that the duration of Shh inhibition might be the deciding factor on PDAC outcomes, as acute or early inhibition might promote tumor growth, as seen in deletion of Shh in genetic models, while targeting Shh in well-established tumors can decrease the tumorigenic potential and enhance drug delivery. Further shedding light on this conflict, Steele et al. found that Hedgehog signaling intersects with CAF heterogeneity in PDAC, as myCAFs have differentially upregulated Shh signaling compared to iCAFs. Acute inhibition of this pathway can change the myCAF/iCAF ratio, transforming the TME into a more immunosuppressive phenotype [79].Clinical trials that attempted to target this pathway in PDAC failed to generate positive results [80,81,82,83,84]. One reason for their failure could be the fact that these trials did not take into account the heterogenous response of CAF populations in PDAC TME to Shh signaling inhibition.

ECM secretion by CAFs can also increase the interstitial pressure in the PDAC TME, which leads to vascular collapse, impeding blood flow, and consequently, drug delivery, and immune cell infiltration. Aiming to ameliorate this hydrostatic pressure through breakdown of hyaluronan, researchers administered hyaluronidase (pegylated recombinant human hyaluronidase 20 (PEGPH20)) into GEMMs of PDAC. This resulted in the depletion of hyaluronan, along with improved vascular diameter and blood flow, which led to increased gemcitabine delivery and tumor response [85,86]. However, clinical translation of this strategy has been disappointing, as two different clinical trials failed to show any benefit of PEGPH20 therapy in combination with standard-of-care chemotherapy in advanced PDAC [87,88,89]. Studies have also evaluated the role of the antihypertensive drug Losartan for decreasing the stromal stress in PDAC. Losartan can attenuate collagen and hyaluronan deposition by CAFs through inhibition of TGF-β signaling in orthotopic PDAC models. This can improve blood flow and drug delivery to tumors, while decreasing the density of activated αSMA^+^ CAFs [90,91]. Interestingly, a single-arm phase II study of total neoadjuvant therapy in the form of a combination of Losartan with radiotherapy and FOLFIRINOX (Leucovorin calcium, 5-Flourouracil, Oxaliplatin, Irinotecan) for locally advanced PDAC showed extremely promising results, with 42/49 (86%) of patients able to undergo attempted resections, and R_0_ resection was achieved in 61% of the cases [92,93].

Instead of targeting the ECM secretory pathways in the CAFs themselves, preclinical studies have also been undertaken to disrupt cancer-cell mediated cues that instigate desmoplastic reaction via CAFs. An example of such a target is the focal adhesion kinases (FAKs). FAKs are non-receptor tyrosine kinases, highly expressed in neoplastic pancreatic cells, which can become activated through ECM receptors, such as integrins, and affect matrix deposition, stiffness, cancer cell invasion, and metastasis [94]. Jiang et al. were able to demonstrate the efficacy of the small molecule FAK inhibitor VS-4718 in reducing tumor growth in the KPC model of PDAC, along with reduced fibrosis and increased infiltration of tumor suppressive cytotoxic CD8 T cells. This therapy was also able to sensitize PDAC to gemcitabine chemotherapy, as well as to anti-PD1 immunotherapy [95]. Other kinases that have been targeted to modulate the ECM in PDAC are the Rho-associated protein kinases (ROCKs). ROCKs are the downstream effectors of the Rho GTPase, which control cell contractility through regulation of the actomyosin complex. In a study by Whatcott et al. ROCKs were found to be upregulated in PDAC, and their inhibition in vitro led to decreased PDAC cell proliferation and migration, as well as reduced PSC activation. In vivo inhibition of ROCKs in the KPC model of PDAC with the small molecule inhibitor fasudil decreased tumor growth and collagen deposition [96]. It was further demonstrated that ROCK inhibition actually disrupts the MMP-mediated ECM remodeling by pancreatic cancer cells, leading to decreased invasive PDAC growth [97].

Lysyl oxidase like-2 (LOXL2) [98], an ECM remodeling enzyme that is upregulated in desmoplastic tumors and helps in maintaining the protumorigenic stroma [99], has also been evaluated as a therapeutic target in PDAC. Inhibition of LOXL2, using the humanized monoclonal antibody Simtuzumab, showed promising results in preclinical models of breast cancer [100], which, combined with the fact that PDAC undergoes extensive desmoplasia, formed the basis of a phase II clinical trial of the combination of Simtuzumab and gemcitabine for metastatic PDAC patients. However, the combination did not provide any survival benefit [101,102]. One limitation of this study is the fact that monoclonal antibodies chiefly inhibit the extracellular LOXL2, but do not affect the activity of intracellular LOXL2, which might play a role in tumor progression as well.

Targeting Metabolic pathways in CAFs: To thrive in the nutrient-deprived environment of PDAC, KRAS-mediated transcriptional machinery drives several metabolic adaptations in the cancer cells, including increased efficiency of glucose utilization [103], diversion of metabolites to biosynthetic pathways, such as the hexosamine pathway and the pentose phosphate shunt [103], and nutrient scavenging through mechanisms, such as autophagy [104] and macropinocytosis [105]. In addition to the cell-specific adaptations, cancer cells can derive metabolic support through the paracrine secretions of their TME neighbors, such as CAFs.

Sousa et al. demonstrated that PSCs secrete alanine in the PDAC TME, which can be utilized by the cancer cells in the TCA cycle to upregulate fatty acid synthesis and serine biosynthetic pathways [106]. Further studies showed that this paracrine–alanine network was dependent on specific transporters, SLC38A2 on the cancer cells and SLC1A4 on the CAFs, and targeting SLC38A2 was sufficient to impede tumor growth [107]. Metabolically reprogrammed cancer cells also rely on non-canonical utilization of glutamine to maintain their redox state [108]. A recent study found that Netrin G1(NetG1), expressed on CAFs, was able to metabolically support PDAC cells through direct transfer of glutamine and glutamate via NetG1-NGL1 (its postsynaptic receptor) interactions [109]. Genetic ablation, as well as pharmacological inhibition of NetG1, were not only able to retard tumor progression, they were also able to alleviate the immunosuppressive TME phenotype through recovery of NK cell-mediated tumor surveillance. Apart from nitrogen–carbon metabolism, CAFs can also support cancer cells through secretion of lipids. PSC-secreted lysophosphatidylcholines can support autotaxin-mediated production of lysophosphatidic acid (LPA) by the PDAC cells. LPA induces cancer cell proliferation, migration, and the AKT signaling pathway; inhibition of autotaxin can abrogate the tumor growth induced by the autotaxin–LPA axis, both in vitro and in vivo [110]. Although, currently, there are no clinical trials underway targeting this metabolic interdependence between PDAC cells and CAFs, this is an avenue that is being extensively explored in preclinical models, and should see translational applications in the near future [111].

Targeting CAF-induced immunosuppression: The PDAC TME has been linked to a delayed wound-healing response, dominated by ECM-depositing fibroblasts, M2 polarized macrophages, myeloid-derived suppressor cells (MDSCs), and Tregs [112]. Indeed, across multiple tumor types, specimens possessing a gene-expression signature similar to the wound-healing response have a worse prognosis [113]. In PDAC, this creates a highly immunosuppressive environment where cytotoxic CD8^+^ and NK T cells are effectively excluded, enabling tumor cells to escape immune surveillance. CAFs have been shown to contribute to this tumorigenic immune TME in multiple studies. The iCAFs identified by Öhlund et al. [40] secrete a number of tumor-promoting cytokines, such as IL-6 and CXCL12. Using co-injection of KPC cancer cells and PSCs orthotopically in a syngeneic mouse model, Garg et al. demonstrated that PSCs secrete CXCL12 in the PDAC TME in an NF-ϏB-dependent manner, which leads to decreased infiltration of cytotoxic CD8^+^ T cells by binding to its receptor CXCR4, resulting in increased tumor growth [114]. This effect could be abrogated through the small molecule CXCL12 inhibitor, AMD3100 [114]. Feig et al. have also shown that inhibition of CXCL12 from FAP^+^ fibroblasts can synergize with PD-L1 blockade to improve immunotherapy against PDAC [21]. Interestingly, CXCL12 has also shown to directly act on pancreatic cancer cells, leading to gemcitabine resistance and increased expression of survival proteins [115]. Data from all of these studies provide strong rationale for clinical trials, with two separate clinical trials targeting the CXCL12–CXCR4 axis already underway [116,117].

IL-6 is another product of iCAFs that supports tumor growth through multiple mechanisms, including gemcitabine resistance [118], increased cancer cell invasion and proliferation [119], and establishment of a metastatic niche in the hepatic parenchyma [120]. In vivo depletion of IL-6 was also able to increase the efficacy of PD-1 blockade in syngeneic PDAC mouse models, as well as to increase survival in the KPC-Brca2 (*Kras*^LSL-G12D^, *Trp53*^LSL-R270H^, *Pdx1-cre*, *Brca2*^F/F^) GEMM [121]. Metabolic reprogramming of the TME through stromal IL-6 can also lead to M2 polarization of macrophages and exclusion of CD8^+^ T cells [122]. A recent study demonstrated an IL33–CXCL3–CXCR2 loop in the PDAC TME. The authors found that IL-6^+^ CAFs secrete IL33, which induced CXCL3 secretion via macrophages. Interestingly, this CXCL3 engages with its receptor CXCR2 on naïve CAFs to convert them into myCAFs, which eventually promote PDAC metastasis [123]. These findings underscore the importance of IL-6 signaling in PDAC TME, and this has been recognized in the form of an active clinical trial employing a combination of IL-6 and PD-1 blockade in PDAC [124]. However, it should be noted that IL-6 can be secreted from other compartments, such as cancer cells and immune cells, and can affect PDAC growth and progression [125], and therefore, systemic depletion of IL-6 will affect CAF-independent pathways in PDAC as well.

Modulating CAF phenotype in the TME: Altering the CAF phenotype in the PDAC TME is also being investigated as a potential therapeutic approach in PDAC. PDAC specimens have high proportions of activated CAFs, characterized by loss of lipid droplets, and expression of markers, such as αSMA. Sherman et al. found that human PDAC CAFs highly express the Vitamin D receptor (VDR), and treatment of these CAFs with calcipotriol can induce a quiescent phenotype, with lipid droplet accumulation and reduced αSMA expression. This treatment also led to decreased secretion of tumor-promoting factors in the CAF conditioned media, while in vivo treatment in a GEMM led to decreased tumor burden and increased survival [126]. This strategy is now being used in a clinical trial where the role of Vitamin D supplementation is being evaluated in both resectable [127,128], as well as unresectable PDAC [129,130].

Another agent that has demonstrated efficacy in induction of CAF quiescence is ATRA (all-trans retinoic acid). ATRA was found to impair the mechanosensory activation, as well as the stromal remodeling capacity of PSCs through a RAR-β (retinoic acid receptor beta) mediated downregulation of MLC2 (myosin light chain 2) contractility [131]. ATRA also showed efficacy in vivo in the KPC model of PDAC, where ATRA treatment led to reduced PSC activation and motility, along with a disrupted Wnt signaling axis in the cancer cells, resulting in decreased tumor growth [132]. ATRA is now being evaluated in conjunction with chemotherapy in PDAC patients [133,134].

Apart from induction of quiescence, there might be opportunities in switching CAF phenotype from an inflammatory one to a myofibroblastic one. Recent studies tend to point towards a more cancer-restraining role for myCAFs, while a cancer-promoting role for the iCAFs. Notably, these phenotypes are reversible in vitro, with myCAFs growing on plastic layers being transformed into iCAFs upon exposure to cancer-conditioned media and vice-versa [106]. In vivo inhibition of JAK signaling, which is essential for IL-1R mediated transformation of CAFs into iCAFs, was able to not only attenuate tumor progression, but also to decrease iCAFs, while increasing myCAFs in tumors from KPC mice [106]. Thus, it seems that attempting to switch CAFs from an iCAF to a myCAF phenotype might be a viable therapeutic approach against PDAC. However, no benefits were observed when the JAK inhibitor ruxolitinib was used in randomized controlled trials for locally advanced or metastatic PDAC [135]. A different approach for this purpose can be antagonism of IL-1R. Preclinical studies have shown that IL-1R or IRAK4 (IL-1 receptor-associated kinase 4) inhibition can lead to decreased tumor growth and reduced stromal content, with abrogation of inflammatory cytokine secretion [52,136]. These findings have led to a clinical trial evaluating the role of the combination of anakinra (IL-1R antagonist) with standard-of-care chemotherapy in patients with PDAC [137].

## 8. Conclusions

The PDAC stroma is a complex and dynamic compendium of cellular and biochemical components, and plays a significant role in tumor progression and metastasis. The stromal compartment has undergone a great deal of investigation in the past decade, thanks to single-cell RNA-sequencing techniques, and as a result, we now understand that CAFs are heterogenous entities, and this has changed the rationale of targeting CAFs in PDAC therapeutics. Now, we are well aware that some CAFs can behave as protumorigenic, while others as antitumorigenic, and therefore, their targeting in solid malignancies demands their thorough characterization. It will be important to delineate whether this heterogeneity results from variable origins of CAFs, or as a consequence of variations in autocrine and paracrine signals in the microenvironment. Current reports suggest that iCAF and apCAF are attractive targets that can be reprogrammed to transform into quiescent CAFs. Still more work is required in terms of their lineage determination and extraction of a good number of viable CAFs from dense ECM, so that their characterization also improves. Moreover, we need to evaluate how durable these subtypes remain under therapeutic pressures, and what their contribution is to therapy resistance in PDAC.

Although there is no approved stroma-targeting therapy yet, a number of clinical trials are underway. Failures of some trials, such as targeting the Hedgehog pathway, have been disappointing, but helped in expanding our horizon of CAF biology. Now, as we understand the CAFs better than ever, it is important to design the clinical trials targeting CAFs carefully, so that CAF-targeting agents with currently available therapeutics will lead to better outcome in PDAC patients.

## Figures and Tables

**Figure 1 ijms-22-13408-f001:**
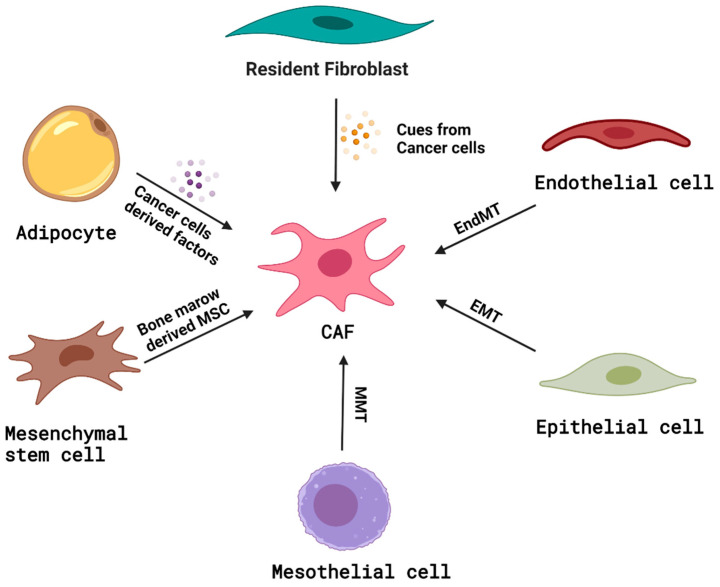
Origin of CAFs. Illustration of the potential cells of origin of CAFs, including resident fibroblasts, endothelial cells, epithelial cells, mesothelial cells, mesenchymal stem cells, and adipocytes with probable mechanisms.

**Figure 2 ijms-22-13408-f002:**
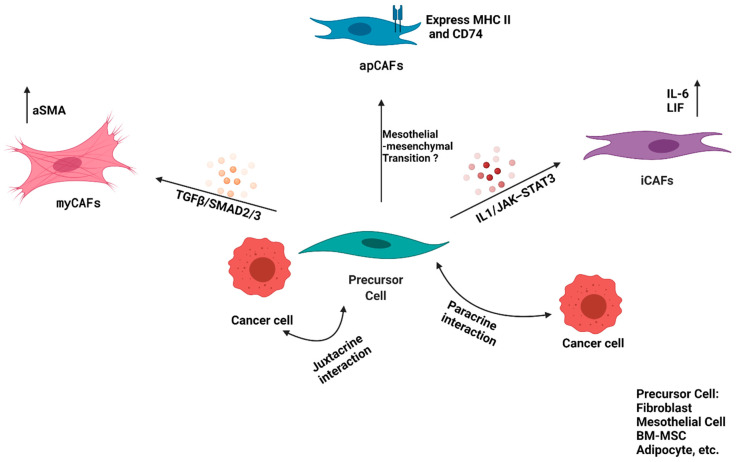
CAF Heterogeneity. Illustration of CAF heterogeneity, showing differentiation of precursor cells, such as resident fibroblast or mesothelial cells, or BM-derived MSCs, etc., into myCAF, iCAF, and apCAF, with reported probable mechanisms.

**Figure 3 ijms-22-13408-f003:**
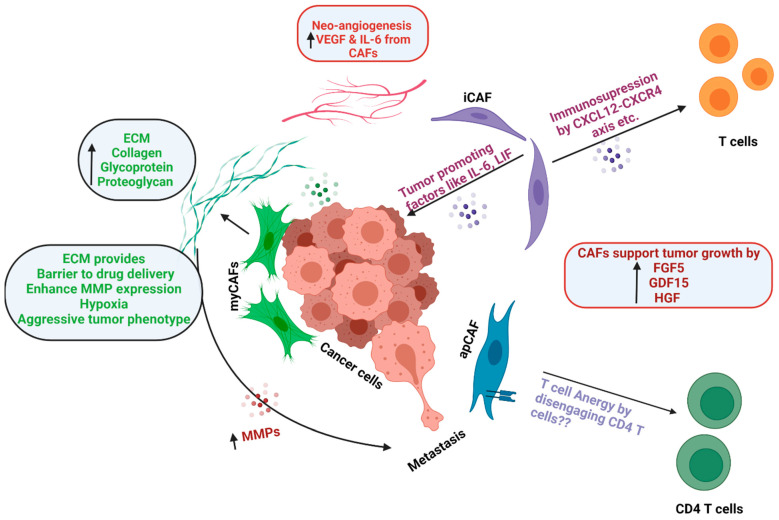
Functions of CAFs. Schematic representation of the functions of CAFs in the tumor microenvironment, such as ECM deposition, immunosuppression, tumor promotion, neoangiogenesis, metastasis, etc.

**Table 1 ijms-22-13408-t001:** Clinical trials targeting CAFs in PDAC.

S. No.	Target	Name of Therapeutic	Rationale Based on Pre-Clinical Studies	Current Status	ClinicalTrials.gov Identifier
1.	Hedgehog Pathway	IPI-926	Inhibition of the Hedgehog Pathway, leading to reduced CAF activation	Phase II was halted due to the early detection of a shorter median overall survival (OS) in the experimental arm, compared to the placebo arm.	NCT01130142
2.	Hedgehog Pathway	Vismodegib		The phase Ib/II randomized clinical trial, evaluating the addition of Vismodegib to gemcitabine, showed no treatment benefit for OS or progression free survival (PFS).	NCT01195415
3.	Hyaluronic acid	PEGPH20	Depletion of stroma by PEGPH20, which may synergize with immunotherapy	Clinical trials failed to show any benefit.	NCT03634332
4.	Angiotensin receptor	Losartan	Attenuation of collagen and hyaluronan deposition by CAFs through inhibition of TGF-β signaling	Encouraging results in locally advanced PDAC	NCT03563248
5.	LOXL2	Simtuzumab	Inhibition of matrix-remodeling enzyme Lysyl oxidase-like 2, an ECM remodeling enzyme	Study completed, and addition of Simtuzumab to gemcitabine did not improve clinical outcomes	NCT01472198
6.	CXCL12-CXCR4 Axis	Olaptesed (NOX-A12)	Modulation of PDAC TME by reducing immunosuppressive factors, as CXCL12 secreted by iCAFs promotes tumorigenesis by reducing CD8 T cell infiltration	Clinical trial ongoing	NCT03168139
7.	CXCL12-CXCR4 Axis	BL-8040 CXCR4 Antagonist		Clinical trial ongoing	NCT02826486
8.	IL-6	Siltuximab	Combination of IL-6 and PD-L1 blockade decreases tumor growth, improves survival, and leads to increased infiltration of effector CD8+ T cells	Clinical trial ongoing	NCT04191421
9.	Vitamin D receptor (VDR)	Paricalcitol (Vitamin D Receptor Agonist)	Modulating signaling in tumor microenvironment. CAFs highly express VDR, and treating them with Vitamin D can induce a quiescent phenotype	Clinical trials ongoing	NCT03520790
					NCT03300921
					NCT02754726
10.	Vitamin D receptor (VDR)	Vitamin D3		Clinical trials ongoing	NCT03472833
11.	Stroma	All Trans Retinoic Acid (ATRA)	Inducing CAF quiescence and decreasing motility, leading to decreased tumor growth through decreased Wnt-β Catenin signaling	Clinical trials ongoing	NCT03307148
12.	IL-1R	Anakinra (IL-1R antagonist)	By switching iCAF to a myCAF phenotype	Clinical trials ongoing	NCT02550327

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
