# Peer review of "Cancer-Associated Fibroblasts in Pancreatic Ductal Adenocarcinoma: An Update on Heterogeneity and Therapeutic Targeting"

_ijms, 2021, doi:10.3390/ijms222413408_

Round 1

Reviewer 1 Report

  1. At first, pancreatic cancer classification included many histological types, e.g. acinar cell carcinoma, serous cystadenocarcinoma, etc. This article only include one of them. Did the title need to be change to pancreatic ductal carcinoma for clarification?
  2. In the introduction line 27, PDAC has only recently reached double digit at 10%. I wonder what the this comes from? According to the “CA Cancer J Clin. 2020 Jan;70(1):7-30.”, there is no PDAC data. Only pancreatic cancer data reported. Also, it is not at 10%. There is variation between 5.5 to 14%. The authors should be more conscious about wording.
  3. Introduction section line 31, current therapeutic strategies is not entirely right. Patient with early pancreatic cancer should receive operation and then consider chemotherapy, or target therapy.
  4. Could the authors clarify why did they pick up histologic types of ductal adenocarcinoma instead of other histology type? It’s not clear in the introduction section. The authors might refer to “Fibroblasts in Pancreatic Ductal Adenocarcinoma: Biological Mechanisms and Therapeutic Targets. Gastroenterology . 2019 May;156(7):2085-2096.”
  5. Page 2 of 19, the authors only mention about one sentence of microbiome. It’s very strange to put it here and the author didn’t make it clearer.
  6. Page 3 of 19 line 98, “extensively studies” authors should cites the most recent study or related review article.
  7. The origin of CAFs page 2 of 19 is not written very clear. Single-cell RNA sequencing analysis of healthy pancreata has shown the presence of distinct fibroblast populations that may represent different precursors of PDAC myofibroblastic CAFs (myCAFs) and inflammatory CAFs (iCAFs). The authors might need to clarify it.
  8. Table one, notable clinical trials targeting CAFs in PDAC. How to define notable?
  9. How to make sure all the therapy are focusing on fibroblasts. Strictly speaking, for example, table 1, IL-6, PD-1 immunotherapy is not focus on fibroblast. Though you can claim that all will affect fibroblass.
  10. Gene while being written should be italic, e.g. KRAS, CDKN2A.
  11. Besides, may English spelling errors.

Author Response

Reviewer I

We thank the Reviewer for evaluating our manuscript and providing valuable insights and critiques. We have tried our best to address all concerns and a point-by-point summary of our modifications is listed below. All the changes in revised manuscript are highlighted using red font. We believe that these modifications have significantly improved our manuscript and addressed the Reviewer’s concerns. 

  1. At first, pancreatic cancer classification included many histological types, e.g. acinar cell carcinoma, serous cystadenocarcinoma, etc. This article only include one of them. Did the title need to be change to pancreatic ductal carcinoma for clarification?

Response:

We agree with the reviewer that pancreatic cancer has many types. Exocrine pancreatic cancer represents approximately 95% of all pancreatic cancer and roughly 85% of primary pancreatic cancer are pancreatic ductal adenocarcinomas (PDACs). However, up to 10% of pancreatic cancer have a rare histotype like acinar cell carcinoma (ACC), adenosquamous carcinoma and other exocrine variants.

Therefore, as per reviewer’s suggestion, we have changed “pancreatic cancer to pancreatic ductal adenocarcinoma” in the title (line 2) of the manuscript.

  1. In the introduction line 27, PDAC has only recently reached double digit at 10%. I wonder what the this comes from? According to the “CA Cancer J Clin. 2020 Jan;70(1):7-30.”, there is no PDAC data. Only pancreatic cancer data reported. Also, it is not at 10%. There is variation between 5.5 to 14%. The authors should be more conscious about wording.

Response:

We thank the reviewer for the comment and have replaced ‘PDAC’ with ‘pancreatic cancer’ to make our statement accurate. We have also added another reference for this statement i.e., “CA CANCER J CLIN 2021;71:7–33” showing the 5-year relative survival rate for pancreatic cancer as 10%.

  1. Introduction section line 31, current therapeutic strategies is not entirely right. Patient with early pancreatic cancer should receive operation and then consider chemotherapy, or target therapy.

Response:

We agree with the reviewer that patient with early pancreatic cancer undergo surgery and then chemotherapy. We wanted to highlight the fact that the systemic therapies available for patients with PDAC attempt to target only the cancer cells, but the outcomes remain dismal despite this standard-of-care therapy. This further underscores the significance of developing therapies targeted at other compartments of the tumor microenvironment. To better communicate this message, we have re-worded our Introduction. We hope this will allay the reviewer’s concerns.

  1. Could the authors clarify why did they pick up histologic types of ductal adenocarcinoma instead of other histology type? It’s not clear in the introduction section. The authors might refer to “Fibroblasts in Pancreatic Ductal Adenocarcinoma: Biological Mechanisms and Therapeutic Targets. Gastroenterology . 2019 May;156(7):2085-2096.”

Response:

We thank the reviewer for the comment.

Pancreatic ductal adenocarcinoma (PDAC) accounts for approximately 85% of all pancreatic cancer cases, has poor prognosis and despite decades of research, the patient outcome is depressing. PDAC has undergone extensive evaluation in context of the stromal biology and CAFs in particular. And that’s why we have discussed the CAFs in context of PDAC only in the current review. The reviewer will see in the revised manuscript that we have modified the title and introduction of the manuscript accordingly.

  1. Page 2 of 19, the authors only mention about one sentence of microbiome. It’s very strange to put it here and the author didn’t make it clearer.

Response:

We appreciate the reviewer for bringing up this point. The microbiome has been well characterized as an important determinant of tumor growth in various solid cancers like colorectal cancer and melanoma, but only recently has been established as an important immunomodulator in the PDAC microenvironment. Evidence from pre-clinical studies suggests that the gut microbiome can alter intra-tumoral adaptive and innate immune system. Another recent study using human specimens has shown that even the intra-tumoral microbiome in pancreatic cancer correlates with survival and immune-cell infiltration. Intra-tumoral fungal pathogen Malassezia has also been shown to affect PDAC progression. Also, while microbiota-stellate cell interaction has been studied in hepatocellular cancer, this is still an unexplored area in PDAC. Based on such studies, we believe that microbiome should also be recognized as an important component of the PDAC microenvironment and this will emerge as a therapeutic target in the PDAC TME in near future. However, to maintain focus, we have removed the reference to the commensal microbiome in the introductory section.

  1. Page 3 of 19 line 98, “extensively studies” authors should cites the most recent study or related review article.

Response:

            We thank the reviewer for the comment and added reference to the text.

  1. The origin of CAFs page 2 of 19 is not written very clear. Single-cell RNA sequencing analysis of healthy pancreata has shown the presence of distinct fibroblast populations that may represent different precursors of PDAC myofibroblastic CAFs (myCAFs) and inflammatory CAFs (iCAFs). The authors might need to clarify it.

Response:

Heterogenous fibroblastic origin of CAFs is indeed indicated by recent studies, as the reviewer pointed out. We had previously discussed this study by Dominguez et al in the “CAF heterogeneity” section. We have now elaborated upon it in the ‘Origin of CAFs’ section instead, as per the reviewer’s suggestion, to discuss the fibroblastic origin of CAFs in more detail. We have also added another recent study by Garcia et al in the same section which describes heterogenous fibroblast populations in the normal pancreata, delineated by Gli1 and Hoxb6 expression, and lineage traced their evolution into CAFs.

  1. Table one, notable clinical trials targeting CAFs in PDAC. How to define notable?

Response:

Our intention was to highlight some recent clinical trials which were based in pre-clinical evidence generated from stroma-targeting studies. However, as the reviewer correctly points out, it might not be appropriate to label them as ‘Notable’ at the expense of other trials that we have not summarized here. Therefore, we have removed this word from the description of Table 1.

  1. How to make sure all the therapy are focusing on fibroblasts. Strictly speaking, for example, table 1, IL-6, PD-1 immunotherapy is not focus on fibroblast. Though you can claim that all will affect fibroblasts.

Response:

We agree with the reviewer that IL6 has pleotropic effects in the PDAC TME and affects multiple compartments including cancer cells, CAFs and immune cells. Also, IL6 secreted from  compartments other than CAFs, such as macrophages or cancer cells, can have tumor promoting roles. However, we feel that it should be mentioned under therapies which are targeting CAFs as some studies suggest that iCAFs are the predominant source of IL-6 in the PDAC TME (Öhlund et al, 2017, Mace et al, 2018)[1, 2]. Therefore, depleting IL-6 would also target numerous effects of iCAFs on PDAC progression. We have acknowledged in the text (519-522) that targeting IL6 will have direct impact on additional pathways which might be independent of CAF signaling.

We have also changed mechanism to “Rational based on Pre-clinical studies” in Table 1 and also elaborated the rational so that the therapies look more fibroblast-centric.

  1. Ohlund, D., et al., Distinct populations of inflammatory fibroblasts and myofibroblasts in pancreatic cancer. J Exp Med, 2017. 214(3): p. 579-596.
  2. Mace, T.A., et al., IL-6 and PD-L1 antibody blockade combination therapy reduces tumour progression in murine models of pancreatic cancer. Gut, 2018. 67(2): p. 320-332.

  1. Gene while being written should be italic, e.g. KRAS, CDKN2A.

Response:

            We thank the reviewer for the comment and italicized the gene names in our manuscript.

  1. Besides, may English spelling errors.

Response:

We apologize the reviewer for the spelling errors. We have gone through the entire manuscript and tried to correct all the errors.

Reviewer 2 Report

In this Review, the Authors provided a detailed and complete report of the literature recently published about the role of CAFs in PDAC. They explored both the origin of CAFs and their heterogeneity and their possible role as targets for therapeutic intervention.

I think this review is well written and I have only minor concerns:

  • I suggest to switch the paragraphs "Functions of CAFs" and "CAFs heterogeneity", in order to make figure 2 easier to understand. In any case figure 2 should be more explanatory.
  • Minor check of the text should be done ( i.e. lane 112,268, and others) and please pay attention to the use of CAF or CAFs.

Author Response

Reviewer II

We thank the reviewer for their compliments regarding our manuscript and their helpful suggestions. As mentioned below, we have revised our manuscript as per the reviewer’s suggestions. All the changes in revised manuscript are highlighted using red font.

  1. I suggest to switch the paragraphs "Functions of CAFs" and "CAFs heterogeneity", in order to make figure 2 easier to understand. In any case figure 2 should be more explanatory.

Response:

As per the reviewer’s suggestion we have switched the two paragraphs. This switching makes Figure 2 as 3 and Figure 3 as 2.

We have also made figure 3 more explanatory. Besides we have also made little modification in figure 2.

  1. Minor check of the text should be done (i.e. lane 112,268, and others) and please pay attention to the use of CAF or CAFs.

Response:

We thank the reviewer for pointing out the errors. We have made the suggested corrections and we have also gone through the entire manuscript and tried to correct all the errors. We have also made corrections pertaining to use of CAFs and CAF.

Round 2

Reviewer 1 Report

English Proof reading should get a credible certification.